# The Impact of RDP Measures on the Rural Development: The Case of Romania

**Ramona Pîrvu \*, Laurențiu Dragomir, Bogdan Budică, Răducu-Ștefan Bratu, Sorin Dinulescu and Lili Țenea**

Department of Economics, Accounting and International Business, University of Craiova, 200585 Craiova, Romania; laurentiu.dragomir@edu.ucv.ro (L.D.); bogdan.budica@edu.ucv.ro (B.B.); raducu.bratu@edu.ucv.ro (R.-Ș.B.); sorin.dinulescu@yahoo.ro (S.D.); office.tenea@gmail.com (L.Ț.)
\* Correspondence: ramona.pirvu@edu.ucv.ro or ramopirvu@gmail.com; Tel.: +40-0722-912-316

**Abstract:** The goal of this study is to analyze the development of rural areas in Romania on the basis of the absorption of both the European non-reimbursable funds and the national funds for the modernization of the infrastructure of villages and communes. The aim of the article is to carry out thorough empirical research on the process of rural development in Romania as a facet of the EU's overall rural development. Thus, in order to obtain relevant results in connection with the pursued goal, we will be using hierarchical cluster analysis to observe the effects of the expenses financed through the National Rural Development Program (NRDP) on the development of rural areas. In accordance with the methodological approach of research, our goal is to give a picture of the way in which Romania's counties (NUTS 3) were clustered at the beginning of the multi-annual planning (2014) as well as at the end of the planning period, in 2020, and to point out a series of practical and concrete aspects generated by the opportunities that the rural areas have had in terms of attracting structural funds. Based on the conducted analysis, we can say that a series of positive aspects can be put in the spotlight as a result of the existence of the European irredeemable funds, which engender positive direct and indirect effects on both the economy and the population's standard of living.

**Keywords:** rural development; Common Agricultural Policy; sustainable development; European funds

## 1. Introduction

The development of the rural areas has caught the eye of an increasing number of specialists [1–3]. The European rural development policy is relevant for the future of rural areas in Europe for three main reasons: first of all, rural development is a response to the "pressure" on the shoulders of European agriculture because rural development currently mobilizes new sources of income to increase an otherwise stagnant agrarian revenue [4,5]. We consider that rural development provides the foundations for rebuilding the eroded economic basis of both the rural economy and agricultural enterprise. Secondly, amongst the key aspects of rural development, the emergence of a new link between agriculture and society comes to the surface [6]. Rural development implies the creation of new products/services, new markets, and new ways of bringing down costs, which in general coincides with the needs and expectations of the society. Thirdly, rural development is also about redefining and reorganizing rural resources. Through rural development the rural resources—land, labor force, nature, ecosystems, animals, plants, craftsmanship, networks, market partners, and urban–rural relations—are remodeled and recombined, as happened, for instance, with the emergence of the alternative food supply chains [7]. In other words, through rural development, new resources are mobilized and combined with the existing ones, ensuring ecological stability and new robust economic mechanisms. The new combinations of resources also allow for new multifunctional enterprises and new networks that link the rural environment with the urban one [8–10].

Over the last few decades, there has been an increasing scientific interest for what can be described as "a new development trajectory" within rural Europe [11,12]. Rural

development tends to be regarded as a politically based strategy with social, economic, and ecological components [13,14] and, at the same time, as a continuous and evolving process which needs constant reevaluation. Murdoch et al. [15] stated more than two decades ago: "sustainability has a broader sense, in that it encompasses the viability of locations and communities on which ultimately, both the preservation of the environment and the economic activity depend. For those concerned with the economic and social development of rural communities, to keep that in mind is crucial".

Rural development is linked with the Common Agricultural Policy, which is one of the most important sectoral policies of the EU. When it was first created, the Common Agricultural Policy aimed at supporting those countries which faced food shortage problems; as a consequence, production-based objectives dominated the agenda [16]. However, since then agriculture has taken a back seat while other problems have risen to the surface, such as food security and environment protection, especially after Agenda 2000 and the outlining of the issues surrounding the future agricultural and rural policy. Hence, because of the extremely changing geopolitical context, as well as the social and economic challenges, the Common Agricultural Policy (CAP) has been forced to constantly modernize. It also had to adapt to the complexity of realities generated by the expansion of the accession process and the doubling of the agricultural population at the European level, as well as the emergence of some new challenges, such as climate change, food security, and the sustainable management of natural resources [17,18].

With the reform of the Common Agricultural Policy, rural development has started to play a growing role within regions with the aim of allowing them to deal with economic, social, and environmental issues at the dawn of the 21st century [19]. At present, rural development has become extremely important for the European community because approximately 60% of EU's population lives in rural areas, which account for 90% of the Union's territory.

The economic diversification of the rural communities, the land management, and the natural resources management are tied up with forestry and agriculture, and for this reason, the rural development policy has currently become a general priority of the EU.

Ever since its appearance, the CAP has had an important significance, reflected also in the EU's budget allotment: 66% of the EU's budget at the beginning of the 1980s and over 37% between 2014 and 2020, while for the present financial framework, the CAP share accounts for 31% of the EU's budget [20]. Initially constructed as a sectoral policy, the CAP currently has a stronger territorial vocation, allowing for possible convergences and overlaps with the objectives of the cohesion [21,22]. Wieliczko, Kurdy's-Kujawska, and Floriańczyk [23] show that the rural policy does not only involve the second pillar of the CAP, but on the contrary, the rural policy is an integral part of several different policies. The latter do not necessarily involve rural development, but they include it amongst their concerns: the fishing policy, the economic and social policy, and the regional development policy (cohesion policy).

In order to fulfill the new objectives of the CAP, two domains of interdependent policies have been formulated and they are considered to be the two pillars of the CAP: (a) the market-oriented agricultural policy (the first pillar) and (b) the rural development policy (the second pillar). This implied a transition from a sectoral to a spatial policy, which assumed a reconnection between the agricultural and the local community, opening up the perspective on the "regionalization" or "territorialization" of the rural policy [24]. Moreover, this meant a switch in the agricultural subsidies, from price support to a more integrated approach which nevertheless sees agriculture as an essential and integrative part of rural areas. The switch in emphasis towards an EU "integrated" rural development was reflected in the reformation proposals of the CAP. Hence, since its creation, the rural development policy has been conceived of as a new field of policy that interacts with other fields of policies, especially the environment policy, the regional policy, etc. [25]. As we showed earlier, the CAP is built around two pillars: the first pillar, which considers the direct payments for agriculture as well as the market measures and accounts for 75.3%, and

the second pillar or, more precisely, the rural development policy, which accounts for 24.4% of the CAP budget [20].

Fulfilling the vertical integration of the rural development policy falls under the responsibility of the national and/or regional authorities because the EU stresses the importance of the subsidiarity principle [26]. The feedback on the implementation process has identified many irregularities and issues regarding its functioning, the relationship between the Commission and the member states, and last but not least, its adoption at the national level. The implementation of the rural development policy is accompanied by different degrees of decentralization and, consequently, by ascending strategies which reflect the different national administrative, organizational, and institutional capacities.

From a rhetorical point of view, the rural development policy is considered a "pluralist" policy for rural areas, which is reflected in the adoption of a large number of extremely diverse objectives, thus confirming the different interpretations and conceptualizations of the rural policy [27]. These objectives mirror the requests of different actors, interest groups, and/or public administrations, as well as the established regional, national, and/or local agendas. Several community texts give birth to or reflect different ideas and approaches with regard to the objectives and content of the rural development policy.

Falling into step with the orientations of the EU policies, the rural development policy includes many important procedures. The analysis of the vertical integration level provides significant proof of an evasive policy with contradictory political objectives and contradictory purposes, which puts together different political measures, organizes political actors with uneven powers, and organizes different actors at the national and regional levels with different institutional and organizational capacities [28].

The general objective of the research focuses on the analysis of the development of rural areas in Romania from the perspective of absorbing European non-refundable funds as well as national funds to modernize the infrastructure of villages and communes and to achieve sustainable development. In this respect, our purpose is to identify the multisectoral impacts of actions directed towards ensuring rural development by taking into account the internal sectoral interconnections as well as the development opportunities of rural areas through the structural funds. We will be using an econometric analysis of the impact of European funds on the development of rural areas in Romania.

The purpose of the carried-out research has a double meaning: (a) to bring theoretical and methodological contributions with regard to the relation between the CAP, rural development, and the overall process of growth and economic development and (b) to deepen the process of rural development in Romania as a facet of the rural development within the EU. The quantitative data collected according to the specified methodology will allow us to identify relevant directions towards the development and diversification of economic activities within the rural space in Romania. To identify directions for the development and diversification of economic activities within the rural space, it is important to understand the relation between the relevant factors and the processes of rural economic development. With the desire to contribute to new knowledge regarding this important topic and by taking as a starting point the results of the preliminary analysis of the field literature, as well as by studying the evolution of the financial allocation of European funds for rural development in Romania, we have decided to explore and measure the economic and social impact of these funds on the rural areas of Romania. Thus, in order to obtain relevant results in connection with the pursued goal, we will be using the hierarchical cluster analysis to observe the effects of the expenses financed through the National Rural Development Program (NRDP) on the development of rural areas.

## 2. Literature Review

The Common Agricultural Policy (CAP) is the exclusive attribute of the European Union, being the first policy elaborated by this supranational integrationist economic organization and regarded as the "the driving force behind the unification of the European space". In essence, the Common Agricultural Policy consists of a "set of principles, methods

and means of action through which the objectives of the European Union in the field of agriculture are achieved" [29].

The Common Agricultural Policy has been and still is one of the basic goals of the European construction and one of the most difficult chapters of the economic integration process.

The CAP can be defined as a set of rules and mechanisms which regulate the production, processing, and trading of agricultural products within the European Union, giving special and increasing attention to rural development [30]. Throughout its evolution the Common Agricultural Policy has been subject to many reforms aiming at attaining new objectives generated both by the EU's expansion and by the deepening of the integration process: improving the competitiveness of the EU's agricultural produce through price reduction; guaranteeing the security and quality of food for consumers; improving the production system by protecting the environment; integrating the components of the environment into the instruments and the goals of the CAP; introducing within the CAP a distinct policy, that is, rural development to guarantee stable revenues and an equal standard of living for the population employed in the field of agriculture; simplifying the legislation; and providing flexibility in the distribution of financial support amongst member states. At present, the CAP puts emphasis on the importance of the environment and on the effects of climate change.

For the period of 2021–2022, a transitory regulation was implemented for the future framework of the CAP strategic plans, that is, the regulation extending most of the norms existing during the period of 2014–2020. The new legislation of the Common Agricultural Policy (CAP), which was formally adopted on 2 December 2021, shall come into effect in all EU member states starting from 1 January 2023.

As for the decision-making process in Romania with respect to the 2014–2020 period, the Ministry of Agriculture and Rural Development implemented the 2014–2020 National Rural Development Program through the Management Authority. For the upcoming period, in line with the goals stipulated in the European Commission's Communication entitled "The future of food and farming", the 2023–2027 National Strategic Plan will aim at attaining the goals of the European Ecological Pact according to the provisions of both the Regulation proposal for the CAP National Strategic Plans and the Regulation proposal concerning the financing, management, and supervision of the CAP [31]. The National Strategic Plan (NSP), which will be the bedrock of the financial allocations from the European funds distributed to Romania for agriculture and rural development for the future 2023–2027 period, was transmitted on 28 February 2022 to the European Commission. The technical and financial implementation of the National Rural Development Program is conducted through the Agency for Financing Rural Investments (AFIR), an institution which is subordinate to the Ministry of Agriculture and Rural Development (MADR).

As we have stated above, the Common Agricultural Policy is focused on several objectives, amongst which is rural development.

If in the beginning the rural development policy overlapped in certain aspects with the Common Agricultural Policy, which was regarded as the main force behind growth and prosperity, today the economic development of the rural environment must be grounded on the changes that emerged due to the progress achieved by society. It also must take into account the new social and economic realities, and last but not least, it must adapt to the new demands of community life.

In this regard, we have tried to provide a general overview of the literature in the field and to point out the chief financing programs of the rural development policy as well as several conceptual approaches and strategic visions of some EU member states with respect to the rural development policy.

Other studies [32,33] described and underlined the importance of the LEADER program in developing the EU's rural economy, and they highlighted its evolution over time and the underlying changes that occurred at each stage in order to stress its ongoing improvement. Wieliczko et al. [23] tried to grasp in their research the incontestable value of

the LEADER program as the bedrock of rural economic development, and they pointed out the necessary measures for increasing life quality.

In our opinion, we cannot overlook the undeniable importance of the Common Agricultural Policy as a driving force behind the overall economic development of the EU's rural environment either. In this respect, the authors Blanco et al. [34] analyzed the financing programs within the European agricultural policy and their decisive role in economic growth and the labor force employment rate.

Castellano-Álvarez et al. [35] talk about the multisectoral approach of the Rural Development Programs. In their opinion, the latter must have in their implementation a multisectoral approach, built on several action points incorporated into a single development strategy. The multisectoral approach of the Rural Development Programs is in tight connection with the economic diversification goal. It implies the implementation of activities in several economic sectors as well as their complete integration into a unique development strategy oriented towards the creation and conservation of the highest possible added value.

Castaño et al. debate in their study upon the concept of *counterfactual analysis* in order to assess the impact of the Rural Development Programs as well as their role in supporting rural communities [36]. The counterfactual analysis was suggested as a way of measuring the impact of the Rural Development Programs (RDPs) over the last few years, although its use in practice has been limited up to the present. Just like any other structural funds, RDPs are subject to evaluations based on results and not forecasts so that their success and impact can be measured in relation to the defined goals. In assessing the impact of the policy, the counterfactual analysis is based on comparing a situation in which a policy was implemented with its counterpoint, that is, when that policy was not applied.

In their paper, Biczkowski et al. refer to the concept of *multifunctionality*, which leads to the rural development of rural areas [37]. Very often, multifunctionality is seen as: (1) a plural activity within the agricultural and industrial systems; (2) a post-productivism paradigm in which agriculture loses its central position in the local economy to the benefit of other ways of exploiting land; (3) within the sustainable development paradigm in which the agricultural production is connected with the socio-economic development of rural areas but also integrated into the whole economy.

Maria de Fátima Oliveira et al. [38] are in support of the idea that innovating the management of the agricultural systems is a key factor which guarantees the adjustment to a new paradigm of the global economy, of the protection of the environment, and of the social demands. The authors feel that stimulating the innovation in the agricultural sector requires a change of the innovation policies at the level of the RDPs in order to preserve natural resources; at the same time, they consider it necessary to intertwine the agricultural priorities and the rural environment with the innovation concepts.

Volkov et al. [39] consider that one of the major flaws in the direct payments scheme is the uneven allotment of resources; that is, there is more for the agricultural sectors of the already developed older member states and less for the developing ones, thus creating disputes between member states.

Ribašauskien et al. [40] agree upon the fact that creating a sustainable agricultural sector involves the stimulation of cooperation activities because they contribute to the social and economic development of farms, farmers, and rural societies.

In their study, Cagliero et al. analyzed the process through which the Common Agricultural Policy (CAP) would be defined for the period of 2023–2027. The implementation model governing the process implies the obligation of each EU member state to elaborate a national strategic plan in order to fulfill those operational actions necessary for exploiting the synergies of the two pillars of the policy. The construction of each plan must be grounded on a proof-based assessment of the needs, and at the same time, this assessment is to be prioritized and planned rigorously in order to create comprehensive, integrated, and achievable interventions [41].

Cagliero et al. observed in their study that the past experiences in the field of rural development polices had brought important changes both at the theoretical and the implementation level. More precisely, these experiences have contributed to the development of the evaluation capacity, and they have granted access to civil society to get involved [42].

In their analysis, Adrian Sadłowski et al. pointed out the weaknesses of the reform of the 2015 Common Agricultural Policy when the public funds were redistributed without taking into consideration the principles of a sustainable agricultural management. Applying some fundamental sustainability criteria for the internalization of the environment externalities would lead to a radical redistribution of payments within the first pillar of the Common Agricultural Policy. From their point of view, a real ecologization of the EU's agricultural policy is an objective still to be attained [43].

The following hypotheses are proposed:

**Hypothesis 1 (H1).** *The economic development of Romania's rural environment is significantly influenced by the opportunities that the rural areas have benefited from as a result of the attraction of structural funds.*

**Hypothesis 2 (H2).** *Significant progress is detected in the development of rural areas—in order to test this hypothesis, we will be using the hierarchical cluster analysis to observe the effects of payments made through the National Rural Development Program (NRDP). The hypotheses will be tested using SPSS.*

Due to the specificity of the research, but also to the scarcity of similar published research, we wanted to conduct a study exploring the impact of RDP measures on rural development in Romania. The hypothesis that the impact is positive can be simplistically defined, but we preferred to use a hierarchical clustering algorithm precisely to try to reveal the existence of potential hidden patterns in the available data. The selected method is a hypothesis-generating, rather than a hypothesis-testing, technique, and therefore we preferred not to define the research hypotheses.

## 3. Research Methodology

Given the nature of the research and the fact that the aim is to identify and analyze the impact of the European funds on the development of rural areas, but also the take into account the constraints generated by the low number of research studies published in the literature, we have decided to use an econometric instrumentarium centered on elaborating some research hypotheses rather than using those instruments specific to testing predetermined hypotheses.

Hence, in order to obtain relevant results in relation to the pursued goal, we will be using the hierarchical cluster analysis to observe the effects of the expenses financed through the National Rural Development Program (NRDP) on the development of rural areas. Cluster analysis is used to group similar variables into groups so that the degree of association between variables is as high as possible if they belong to the same group and as low as possible if they belong to different groups. Cluster analysis is mainly used to reveal hidden structures in the data, but without providing detailed explanations or causal interpretations. In this manner, cluster analysis provides researchers with a distinct way of approaching and interpreting the variables analyzed.

The hierarchical clustering algorithm, in addition to dividing the variables into clusters, also constructs the hierarchy of the distance between the variables, describes how clusters are formed, and shows how different these clusters are from each other. It is a hypothesis-generating, rather than a hypothesis-testing, technique. This is also a reason for this research: to offer a different point of view from the existing literature and to open new research directions.

In accordance with the methodological approach of research, our goal is to give a picture of the way in which Romania's counties (NUTS 3) were clustered at the beginning

of the multi-annual planning (2014) as well as at the end of the planning period, that is, 2020. Having analyzed the payments made through the national rural development plan (NRDP), one can notice that the first payments of the 2014–2020 multi-annual financial framework were made in 2015, but in the same year, the last payments of the previous program (NRDP 2007–2013) were also made, due to the extension of the period in which the request was submitted for the reimbursement of the expenses regarding the projects carried out during the specific period. As a consequence, in order to isolate the effect generated by the remaining payments related to the 2007–2013 NRDP, and at the same time grasp the effect of the 2014–2020 NRDP, we have decided to conduct our analysis based on the payments made in 2016.

In order to explore the impact of the European funds on rural development in Romania, we have decided to analyze the following variables clustered around the counties (NUTS 3) for the years 2016 and 2020: payments made through the NRDP, GDP/county, number of employees, length of modernized public roads (district and rural roads), fleet size of tractors and agricultural machinery, sown areas of main crops, and agricultural output—grains, as well as the working population in agriculture. The data used for the analysis have been collected from several available public sources, such as the Agency for Financing Rural Investments, the National Institute of Statistics, and Eurostat (Table 1).

**Table 1.** Variables selected for analysis.

| Variable | Description | U.M. |
|---|---|---|
| NRDP | Payments made from the 2014–2020 NRDP budget | Thousands of euros |
| GDP | GDP at NUTS 3 level | Millions of euros |
| EMPLOYEES | Number of employees at the end of the year, NACE Rev. 2—agriculture, forestry, fishing | Persons |
| ROADS | Length of modernized public roads (district and rural roads) | Kilometers |
| MACHINES | Fleet of tractors and main agricultural machinery | Number |
| AREAS | Sown area of main crops | Hectares |
| PRODAGR | Agricultural output—grains | Tons |
| POPAGR | Working population employed in different sectors at the level of NACE Rev. 2—agriculture, forestry, fishing | Persons |

Source: calculations made by the authors.

Given the fact that we aimed at comparing the evolution registered between the beginning and the end of the 2014–2020 financial planning period and because statistical data were not available for all of the 2020 variables, we decided to use the value extrapolation method for the year 2019 in order to estimate the values missing for the year 2020.

The extrapolation of missing values was carried out based on the data published for the interval of 2010–2019, for better precision and to minimize the extrapolation errors. For the calculation of the values of the 2020 variables, we used the FORECAST.ETS function from MS Excel (Winston, 2019; Held et al., 2018). Using this method we estimated the 2020 values for the variables GDP, EMPLOYEES, and POP_AGR.

In order to capture the capacity of the parties involved at the level of each county to generate rural development projects and to attract European funds, the variable NRDP.2020 reflects the cumulated value of all payments between 2015 and 2020 made through the NRDP for rural development projects.

Moreover, given that the results of the analysis might potentially be corrupted due to the lack of relevance of the variable values, we decided to drop from our analysis the city of Bucharest and the neighboring Ilfov County. The main characteristics of the analyzed variables are described in Tables 2 and 3.

**Table 2.** Variable descriptive statistics for the year 2016.

| Variable | N | Minimum | Maximum | Average | Std. Deviation |
|---|---|---|---|---|---|
| NRDP | 39 | 10,796.37 | 57,505.4 | 24,856.435 | 12,719.732 |
| GDP | 39 | 1061.99 | 7981.9 | 2963.6026 | 1802.6182 |
| EMPLOYEES | 39 | 1341 | 5198 | 2909.1795 | 1008.4365 |
| ROADS | 39 | 74 | 1174 | 423.79487 | 260.22573 |
| MACHINES | 39 | 2674 | 10,747 | 5020.6923 | 1945.7251 |
| AREAS | 39 | 52,101 | 493,159 | 202,816.95 | 125,474.2 |
| PRODAGR | 39 | 73,979 | 1,327,033 | 505,730.79 | 369,543.35 |
| POPAGR | 39 | 17,500 | 75,600 | 42,248.718 | 14,466.984 |

Source: calculations made by the authors.

**Table 3.** Variable descriptive statistics for the year 2020.

| Variable | N | Minimum | Maximum | Average | Std. Deviation |
|---|---|---|---|---|---|
| NRDP | 39 | 29,302.23 | 170,254.02 | 70,577.30 | 36,479.36 |
| GDP | 39 | 1464.41 | 10,865.13 | 3835.95 | 2466.94 |
| EMPLOYEES | 39 | 1585 | 6026 | 3225.59 | 1070.65 |
| ROADS | 39 | 63 | 1236 | 551.49 | 313.39 |
| MACHINES | 39 | 2148 | 11,996 | 5727.05 | 2428.01 |
| AREAS | 39 | 51,491 | 488,275 | 199,505.92 | 119,904.06 |
| PRODAGR | 39 | 68,369 | 1,325,852 | 423,793.28 | 316,544.82 |
| POPAGR | 39 | 17,500 | 72,200 | 40,748.72 | 13,090.17 |

Source: calculations made by the authors.

In order to be able to use the hierarchical cluster analysis in the way we planned, it is necessary to look at the variables' values to obtain information about the type of distribution, with the goal being to check for the existence of a normal distribution. In this respect, the literature in the field recommends the use of the Kolmogorov–Smirnov test, together with the Shapiro–Wilk test [44,45].

Considering that for the year 2016, the county of Timiş reported a much bigger agricultural output than the other analyzed counties, thus resulting in an extreme value; this county was eliminated from the 2016 analysis. Similarly, in 2020 the county of Dolj registered a much higher agricultural output than the other counties of the country; so, as a consequence, the county was also eliminated from the 2020 analysis due to the fear of extreme values which might have influenced the obtained results.

As for the interpretation of the Kolmogorov–Smirnov and the Shapiro–Wilk test results, we know that there is a correspondence between a large dispersion of values and a small *p*-value. A small *p*-value (typically $\leq 0.05$) indicates strong evidence against the null hypothesis so that you reject the null hypothesis. On the other hand, a large *p*-value (>0.05) indicates that the distribution of data is normal, and this fact is confirmed for all analyzed variables. The results obtained for the years 2016 and 2020 are summarized in Tables 4 and 5.

**Table 4.** Variable normality tests for the year 2016.

| Variables | Kolmogorov–Smirnov [a] | | | Shapiro–Wilk | | |
|---|---|---|---|---|---|---|
| | Statistic [b] | Df [b] | Sig. [b] | Statistic | df | Sig. |
| NRDP.2016 | 0.216 | 39 | 0.110 | 0.849 | 39 | 0.090 |
| GDP.2016 | 0.161 | 39 | 0.072 | 0.824 | 39 | 0.098 |
| EMPLOYEES.2016 | 0.108 | 39 | 0.200 * | 0.960 | 39 | 0.184 |
| ROADS.2016 | 0.113 | 39 | 0.200 * | 0.930 | 39 | 0.017 |
| MACHINES.2016 | 0.151 | 39 | 0.125 | 0.898 | 39 | 0.102 |
| AREAS.2016 | 0.155 | 39 | 0.089 | 0.911 | 39 | 0.085 |
| PRODAGR.2016 | 0.150 | 39 | 0.077 | 0.893 | 39 | 0.081 |
| POPAGR.2016 | 0.117 | 39 | 0.200 * | 0.970 | 39 | 0.367 |

Source: calculations made by the authors. * This is a lower bound of the true significance $p < 0.05$, [a] Lilliefors Significance Correction; [b] "Statistic" is the Shapiro–Wilk test statistic ("W") (Shapiro and Wilk, 1965); "df" stands for degrees of freedom and "Sig." is the p-value (the evidence against a null hypothesis).

**Table 5.** Variable normality tests for the year 2020.

| Variables | Kolmogorov–Smirnov [a] | | | Shapiro–Wilk | | |
|---|---|---|---|---|---|---|
| | Statistic | df | Sig. | Statistic | df | Sig. |
| NRDP.2020 | 0.185 | 39 | 0.092 | 0.880 | 39 | 0.111 |
| GDP.2020 | 0.209 | 39 | 0.100 | 0.807 | 39 | 0.090 |
| EMPLOYEES.2020 | 0.123 | 39 | 0.141 | 0.958 | 39 | 0.152 |
| ROADS.2020 | 0.083 | 39 | 0.200 * | 0.961 | 39 | 0.193 |
| MACHINES.2020 | 0.082 | 39 | 0.200 * | 0.949 | 39 | 0.075 |
| AREAS.2020 | 0.118 | 39 | 0.182 | 0.930 | 39 | 0.077 |
| PRODAGR.2020 | 0.198 | 39 | 0.081 | 0.868 | 39 | 0.065 |
| POPAGR.2020 | 0.114 | 39 | 0.200 * | 0.977 | 39 | 0.589 |

Source: calculations made by the authors. * This is a lower bound of the true significance. $p < 0.05$. [a] Lilliefors Significance Correction.

In order to prepare the data for the cluster analysis, it is necessary to generate the proximity matrix ($W = \|w_{ij}\|_{i=\overline{1,n}, j=\overline{1,n}}$). For the generation of clusters, we used the Euclidean distance (Everitt et al., 2011), according to the Relation (1):

$$W = \|w_{ij}\|_{i=\overline{1,n}, j=\overline{1,n}}, \quad w_{ij} = \sqrt{\sum_{i=1}^{n}(z_{ik} - z_{ij})^2}, \; j = \overline{1,m}, \; k = \overline{1,m} \; j \neq i, \; k \neq i, \; w_{ii} = 0 \tag{1}$$

Next, for finding out the distance between the clusters, we used the Ward method (Ward, 1963), according to the Relation (2):

$$\Delta(A, B) = \sum_{i \in A \cup B} \|x_i - m_{A \cup B}\|^2 - \sum_{i \in A} \|x_i - m_A\|^2 - \sum_{i \in B} \|x_i - m_{\mathbf{B}}\|^2 - \frac{n_{A \cap B}}{n_{A \cup B}} \|m_A - m_B\|^2 \tag{2}$$

For the testing of the significance of the 8 variables pertaining to the clusters, we used Levene's Test whose null hypothesis refers to the fact that the dispersions of the variables do not vary, and we made the calculation according to the Relation (3):

$$H_{0\_1} : \sigma_1^2 = \sigma_2^2 = \sigma_3^2 = \ldots = \sigma_r^2 \tag{3}$$

If the null hypothesis is accepted, then the analysis of variance (ANOVA) can be used for testing the statistical significance of the average values of the analyzed variables, which are grouped into clusters. If the null hypothesis $H_{0\_1}$ is rejected, instead of the analysis of

variance (ANOVA), the Welch or Brown–Forsythe tests can be used, whose null hypothesis $H_{0\_2}$ is given by the Relation (5):

$$H_{0\_1} : m_{ki} = m_{kj}; \; i \neq j \tag{4}$$

In Equation (4), k represents the number of variables ($k = \overline{1,8}$), while $i$ and $j$ represent the numbers of the clusters, ranging from 1 to the maximum number of clusters.

For the processing of the data, the software packages SPSS Statistics (George and Mallery, 2018) and MS Excel (Winston, 2019; Held et al., 2018) were used.

*Model and Method*

The first step in applying the hierarchical cluster analysis method after the preliminary inspection of the available data is to determine both the cluster formation algorithm and the dendrogram. The cluster agglomeration table and the cluster dendrogram for 2016 can be observed in Appendices A and B.

By analyzing the 2016 cluster dendrogram, one can notice the delimitation of 4 clearly determined clusters. The distribution of the analyzed counties within the 4 clusters is shown in Table 6.

**Table 6.** Clusters structure for 2016.

| Cluster | Counties |
|---|---|
| Cluster 2016-1 | Alba, Arges, Bistriţa-Năsăud, Braşov, Caraş-Severin, Cluj, Covasna, Dâmboviţa, Gorj, Harghita, Hunedoara, Maramureş, Sălaj, Sibiu, Suceava, Vâlcea, Vrancea |
| Cluster 2016-2 | Arad, Brăila, Călăraşi, Constanţa, Dolj, Ialomiţa, Teleorman |
| Cluster 2016-3 | Bacău, Botoşani, Iaşi, Mehedinţi, Mureş, Neamţ, Prahova, Vaslui |
| Cluster 2016-4 | Bihor, Buzău, Galaţi, Giurgiu, Olt, Satu Mare, Tulcea |

Source: calculations made by the authors.

As was mentioned in the research methodology, the statistical significance testing of the 8 variables pertaining to the identified clusters will be conducted using the analysis of variance (ANOVA). In order to use this method, it is necessary for the dispersions of the variables to satisfy the dispersion homogeneity test according to the above-mentioned Relation (3), as the null hypothesis implies that the dispersions of variables do not differ significantly. The results of the variance homogeneity tests are shown in Appendix C.

Following the analysis of the homogeneity test results for the variables of the year 2016, the conclusion is that we will reject the assumed null hypothesis given the values higher than the significance level (more precisely, 0.05). This fact suggests that we will not be able to use the analysis of variance (ANOVA). However, according to the literature in the field, as an alternative we can rely on using the Welch and Brown–Forsythe tests, whose null hypothesis implies that the average of variables does not differ significantly. The results of the two tests are presented in Table 7.

By interpreting these findings, we can conclude that we can reject the null hypothesis (for a significance level of 0.05), which implies accepting the alternative hypothesis; that is to say, the average values of the analyzed variables differ significantly from a statistical point of view; therefore, the analysis of variance (ANOVA) is allowed and valid. The results of the analysis of variance (ANOVA) are shown in Table 8.

In order to continue our investigation, that is, to determine the impact of the European funds on the development of rural areas, we will repeat the steps taken to identify and validate the 2016 clusters for the 2020 variables so that we may extract relevant information for our research by comparing the obtained results. The agglomeration table and the clusters dendrogram for 2020 are presented in Appendices E and F.

By analyzing the dendrogram of clusters for the year 2020, we can see the delimitation of 4 clearly determined clusters, which have a different structure than the clusters determined for 2016. The distribution of the analyzed counties within the 4 clusters is presented in Table 9.

**Table 7.** Welch and Brown–Forsythe test results for 2016.

|  |  | Statistic [a] | df1 | df2 | Sig. |
|---|---|---|---|---|---|
| NRDP.2016 | Welch | 2.685 | 3 | 14.066 | 0.037 |
|  | Brown–Forsythe | 1.136 | 3 | 16.354 | 0.034 |
| GDP.2016 | Welch | 0.977 | 3 | 15.194 | 0.029 |
|  | Brown–Forsythe | 0.471 | 3 | 20.546 | 0.006 |
| EMPLOYEES.2016 | Welch | 14.089 | 3 | 15.248 | 0.000 |
|  | Brown–Forsythe | 11.352 | 3 | 25.646 | 0.000 |
| ROADS.2016 | Welch | 1.072 | 3 | 16.178 | 0.039 |
|  | Brown–Forsythe | 0.854 | 3 | 28.766 | 0.047 |
| MACHINES.2016 | Welch | 1.453 | 3 | 14.045 | 0.027 |
|  | Brown–Forsythe | 0.823 | 3 | 17.087 | 0.049 |
| AREAS.2016 | Welch | 61.176 | 3 | 13.554 | 0.000 |
|  | Brown–Forsythe | 64.908 | 3 | 22.063 | 0.000 |
| PRODAGR.2016 | Welch | 133.078 | 3 | 15.041 | 0.000 |
|  | Brown–Forsythe | 206.850 | 3 | 20.066 | 0.000 |
| POPAGR.2016 | Welch | 2.189 | 3 | 15.089 | 0.032 |
|  | Brown–Forsythe | 1.695 | 3 | 22.300 | 0.017 |

Source: calculations made by the authors. [a] Asymptotically F distributed.

**Table 8.** Variance analysis results (ANOVA) for 2016.

|  |  | Sum of Squares | df | Mean Square | F | Sig. |
|---|---|---|---|---|---|---|
| NRDP.2016 | Between Groups | 638,585,849.659 | 3 | 212,861,949.886 | 1.352 | 0.023 |
|  | Within Groups | 5,509,493,929.551 | 35 | 157,414,112.273 |  |  |
|  | Total | 6,148,079,779.210 | 38 |  |  |  |
| GDP.2016 | Between Groups | 4,962,199.337 | 3 | 1,654,066.446 | 0.488 | 0.033 |
|  | Within Groups | 118,516,230.009 | 35 | 3,386,178.000 |  |  |
|  | Total | 123,478,429.346 | 38 |  |  |  |
| EMPLOYEES.2016 | Between Groups | 18,858,308.231 | 3 | 6,286,102.744 | 11.120 | 0.000 |
|  | Within Groups | 19,785,569.513 | 35 | 565,301.986 |  |  |
|  | Total | 38,643,877.744 | 38 |  |  |  |
| ROADS.2016 | Between Groups | 140,128.652 | 3 | 46,709.551 | 0.672 | 0.045 |
|  | Within Groups | 2,433,133.707 | 35 | 69,518.106 |  |  |
|  | Total | 2,573,262.359 | 38 |  |  |  |
| MACHINES.2016 | Between Groups | 11,168,341.047 | 3 | 3,722,780.349 | 0.982 | 0.041 |
|  | Within Groups | 132,693,813.261 | 35 | 3,791,251.807 |  |  |
|  | Total | 143,862,154.308 | 38 |  |  |  |
| AREAS.2016 | Between Groups | 519,380,391,203.086 | 3 | 173,126,797,067.695 | 76.815 | 0.000 |
|  | Within Groups | 78,883,039,868.811 | 35 | 2,253,801,139.109 |  |  |
|  | Total | 598,263,431,071.897 | 38 |  |  |  |
| PRODAGR.2016 | Between Groups | 4,904,411,560,751.960 | 3 | 1,634,803,853,583.990 | 200.797 | 0.000 |
|  | Within Groups | 284,955,477,214.396 | 35 | 8,141,585,063.268 |  |  |
|  | Total | 5,189,367,037,966.360 | 38 |  |  |  |

**Table 8.** *Cont.*

|  |  | Sum of Squares | df | Mean Square | F | Sig. |
|---|---|---|---|---|---|---|
| POPAGR.2016 | Between Groups | 999,384,452.704 | 3 | 333,128,150.901 | 1.677 | 0.019 |
|  | Within Groups | 6,953,772,983.193 | 35 | 198,679,228.091 |  |  |
|  | Total | 7,953,157,435.897 | 38 |  |  |  |

Source: calculations made by the authors.

**Table 9.** Clusters structure for 2020.

| Cluster | Counties |
|---|---|
| Cluster 2020-1 | Alba, Arges, Bacău, Buzău, Cluj, Constanţa, Dâmboviţa, Galaţi, Gorj, Iaşi, Mehedinţi, Neamţ, Prahova, Suceava, Tulcea, Vaslui |
| Cluster 2020-2 | Arad, Bihor, Olt, Teleorman, Timiş |
| Cluster 2020-3 | Bistriţa-Năsăud, Brăila, Călăraşi, Covasna, Harghita, Hunedoara, Maramureş, Satu Mare, Sibiu, Vâlcea, Vrancea |
| Cluster 2020-4 | Botoşani, Braşov, Caraş-Severin, Giurgiu, Ialomiţa, Mureş, Sălaj |

Source: calculations made by the authors.

The statistical significance testing of the 8 variables pertaining to the identified clusters is based on the analysis of variance (ANOVA), which implies that the dispersions of the variables must satisfy the dispersion homogeneity test according to the above-mentioned Relation (3) as the null hypothesis suggests that the dispersions of variables do not differ significantly. The results of the variance homogeneity tests are presented in Appendix F.

Following the analysis of the homogeneity test results for the variables of the year 2020, the conclusion is that we will reject the proposed null hypothesis, given that the significance values are higher than the standard level of 0.05. According to the literature in the field, as an alternative we can rely on using the Welch and Brown–Forsythe tests, whose null hypothesis implies that the average of variables does not differ significantly. The results of the two tests are presented in Table 10.

**Table 10.** Welch and Brown–Forsythe test results for 2020.

|  |  | Statistic [a] | df1 | df2 | Sig. |
|---|---|---|---|---|---|
| NRDP.2020 | Welch | 2.071 | 3 | 13.118 | 0.035 |
|  | Brown–Forsythe | 3.394 | 3 | 15.427 | 0.045 |
| GDP.2020 | Welch | 2.967 | 3 | 13.620 | 0.029 |
|  | Brown–Forsythe | 2.400 | 3 | 13.930 | 0.042 |
| EMPLOYEES.2020 | Welch | 3.652 | 3 | 12.359 | 0.043 |
|  | Brown–Forsythe | 3.229 | 3 | 17.318 | 0.048 |
| ROADS.2020 | Welch | 2.093 | 3 | 14.777 | 0.045 |
|  | Brown–Forsythe | 1.811 | 3 | 31.559 | 0.035 |
| MACHINES.2020 | Welch | 4.460 | 3 | 12.671 | 0.024 |
|  | Brown–Forsythe | 5.476 | 3 | 18.626 | 0.007 |
| AREAS.2020 | Welch | 85.012 | 3 | 11.639 | 0.000 |
|  | Brown–Forsythe | 32.485 | 3 | 23.528 | 0.000 |
| PRODAGR.2020 | Welch | 126.338 | 3 | 11.631 | 0.000 |
|  | Brown–Forsythe | 121.198 | 3 | 7.941 | 0.000 |
| POPAGR.2020 | Welch | 5.821 | 3 | 13.899 | 0.009 |
|  | Brown–Forsythe | 6.533 | 3 | 26.243 | 0.002 |

Source: calculations made by the authors. [a] Asymptotically F distributed.

By interpreting these findings, we can draw the conclusion that we can reject the null hypothesis (for a significance level of 0.05), which implies accepting the alternative hypothesis; that is to say, the average values of the analyzed variables differ significantly from a statistical point of view; therefore, the analysis of variance (ANOVA) is allowed and valid. The results of the analysis of variance (ANOVA) are shown in Table 11.

**Table 11.** Variance analysis results (ANOVA) for 2020.

|  |  | Sum of Squares | df | Mean Square | F | Sig. |
|---|---|---|---|---|---|---|
| NRDP.2020 | Between Groups | 12,225,347,448.124 | 3 | 4,075,115,816.041 | 3.720 | 0.020 |
|  | Within Groups | 38,342,900,485.986 | 35 | 1,095,511,442.457 |  |  |
|  | Total | 50,568,247,934.110 | 38 |  |  |  |
| GDP.2020 | Between Groups | 37,513,106.235 | 3 | 12,504,368.745 | 2.259 | 0.032 |
|  | Within Groups | 193,747,868.615 | 35 | 5,535,653.389 |  |  |
|  | Total | 231,260,974.850 | 38 |  |  |  |
| EMPLOYEES.2020 | Between Groups | 10,688,409.214 | 3 | 3,562,803.071 | 3.794 | 0.019 |
|  | Within Groups | 32,870,416.222 | 35 | 939,154.749 |  |  |
|  | Total | 43,558,825.436 | 38 |  |  |  |
| ROADS.2020 | Between Groups | 420,889.996 | 3 | 140,296.665 | 1.483 | 0.036 |
|  | Within Groups | 3,311,271.748 | 35 | 94,607.764 |  |  |
|  | Total | 3,732,161.744 | 38 |  |  |  |
| MACHINES.2020 | Between Groups | 76,911,654.751 | 3 | 25,637,218.250 | 6.100 | 0.002 |
|  | Within Groups | 147,107,269.147 | 35 | 4,203,064.833 |  |  |
|  | Total | 224,018,923.897 | 38 |  |  |  |
| AREAS.2020 | Between Groups | 368,797,953,128.567 | 3 | 122,932,651,042.856 | 24.237 | 0.000 |
|  | Within Groups | 177,527,380,324.203 | 35 | 5,072,210,866.406 |  |  |
|  | Total | 546,325,333,452.769 | 38 |  |  |  |
| PRODAGR.2020 | Between Groups | 3,595,169,736,073.230 | 3 | 1,198,389,912,024.410 | 197.425 | 0.000 |
|  | Within Groups | 212,453,941,304.665 | 35 | 6,070,112,608.705 |  |  |
|  | Total | 3,807,623,677,377.900 | 38 |  |  |  |
| POPAGR.2020 | Between Groups | 2,156,503,877.456 | 3 | 718,834,625.819 | 5.777 | 0.003 |
|  | Within Groups | 4,354,893,558.442 | 35 | 124,425,530.241 |  |  |
|  | Total | 6,511,397,435.897 | 38 |  |  |  |

Source: calculations made by the authors.

## 4. Empirical Results and Discussion

Given that we were able to determine clusters which were relevant from a statistical point of view for both analyzed periods, 2016 and 2020, respectively, we can assess the characteristics of each cluster based on the variables included in the analysis.

According to the described research methodology, in 2016 there were four significant clusters identified; their characteristics are presented in Table 12.

In the cluster 2016-1, 17 counties were grouped, namely Alba, Arges, Bistriţa-Năsăud, Braşov, Caraş-Severin, Cluj, Covasna, Dâmboviţa, Gorj, Harghita, Hunedoara, Maramureş, Sălaj, Sibiu, Suceava, Vâlcea, and Vrancea.

The main characteristics of the counties grouped into the cluster 2016-1 are given by the highest average values recorded amongst the four clusters for the variables Length of modernized public road-ROADS.2016 (471.59 km of modernized district and rural roads against the national average of 423.79 km in 2016) and Fleet size of tractors and agricultural

machinery-MACHINES.2016 (5190.41 tractors and agricultural machinery against the national average of 5020.69 tractors and agricultural machinery in 2016). Moreover, the counties grouped into this cluster are the ones reporting the lowest average values amongst the clusters analyzed for the variables EMPLOYEES.2016 (2349.94 employees in the field of agriculture against the national average of 2909.18 employees in 2016) and Sown areas of main crops-AREAS.2016 (95,525.65 ha of sown areas against the 2016 national average of 2028.95 ha), alongside Working population in agriculture-POPAGR.2016 (26,994.12 workers in agriculture against the 2016 national average of 42,248.72 workers).

**Table 12.** Cluster characteristics for 2016.

| Cluster | No. of Counties | Variables | Minimum | Maximum | Mean | Variance |
|---|---|---|---|---|---|---|
| Cluster 2016-1 | 17 | NRDP.2016 | 13,952.14 | 46,220.68 | 25,026.91 | 10,741.63 |
| | | GDP.2016 | 1242.80 | 7981.90 | 3041.32 | 1736.56 |
| | | EMPLOYEES.2016 | 1434 | 3979 | 2349.94 | 749.56 |
| | | ROADS.2016 | 85 | 1174 | 471.59 | 318.27 |
| | | MACHINES.2016 | 2758 | 8842 | 5190.41 | 1626.03 |
| | | AREAS.2016 | 52,101 | 176,356 | 95,525.65 | 38,518.06 |
| | | PRODAGR.2016 | 73,979 | 333,948 | 194,178.47 | 95,806.99 |
| | | POPAGR.2016 | 17,500 | 74,300 | 36,994.12 | 14,625.13 |
| Cluster 2016-2 | 7 | NRDP.2016 | 10,796.37 | 55,783.99 | 27,487.60 | 17,230.89 |
| | | GDP.2016 | 1344.54 | 7664.01 | 3108.89 | 2287.69 |
| | | EMPLOYEES.2016 | 3240 | 5198 | 4298.43 | 597.21 |
| | | ROADS.2016 | 132 | 760 | 426.86 | 260.06 |
| | | MACHINES.2016 | 2816 | 7819 | 5354.43 | 2122.11 |
| | | AREAS.2016 | 349,927 | 493,159 | 404,766.00 | 58,983.04 |
| | | PRODAGR.2016 | 965,479 | 1,327,033 | 1,162,542.71 | 116,165.95 |
| | | POPAGR.2016 | 28,300 | 75,600 | 44,700.00 | 17,273.68 |
| Cluster 2016-3 | 8 | NRDP.2016 | 11,523.22 | 26,118.90 | 17,723.26 | 4967.32 |
| | | GDP.2016 | 1243.67 | 7614.65 | 3311.31 | 2184.33 |
| | | EMPLOYEES.2016 | 1341 | 3863 | 2854.00 | 901.71 |
| | | ROADS.2016 | 74 | 508 | 311.63 | 161.60 |
| | | MACHINES.2016 | 2674 | 6857 | 3986.00 | 1336.99 |
| | | AREAS.2016 | 142,824 | 268,257 | 190,500.25 | 48,136.62 |
| | | PRODAGR.2016 | 391,524 | 529,415 | 443,773.63 | 46,576.28 |
| | | POPAGR.2016 | 33,900 | 65,200 | 49,925.00 | 9320.91 |
| Cluster 2016-4 | 7 | NRDP.2016 | 14,016.70 | 57,505.40 | 29,963.47 | 16,878.15 |
| | | GDP.2016 | 1061.99 | 3900.84 | 2232.21 | 954.45 |
| | | EMPLOYEES.2016 | 2232 | 4109 | 2941.14 | 702.92 |
| | | ROADS.2016 | 226 | 719 | 432.86 | 193.14 |
| | | MACHINES.2016 | 2820 | 10,747 | 5457.29 | 2911.39 |
| | | AREAS.2016 | 202,670 | 381,038 | 275,508.71 | 54,849.57 |
| | | PRODAGR.2016 | 588,780 | 812,915 | 676,354.14 | 83,605.42 |
| | | POPAGR.2016 | 22,400 | 61,300 | 43,785.71 | 13,741.84 |

Source: calculations made by the authors.

In the cluster 2016-2, seven counties were distributed: Arad, Brăila, Călăraşi, Constanţa, Dolj, Ialomiţa and Teleorman. Specific to the counties grouped into this cluster is the fact that here are the highest average values of the variable Sown areas of main crops-AREAS.2016 (404,766.00 ha sown areas against the 2016 national average of 202,816.95 ha of sown areas) and of the variable Agricultural output—grains-PRODAGR.2016 (1,162,542,71 tons

of grain production against the 2016 national average of 505,730.79 tons of grain production) as well as the highest average value of the variable EMPLOYEES.2016 (4298.43 employees in agriculture against the 2016 national average of 2909.18 employees in agriculture).

In other words, we might as well characterize the cluster 2016-2 as the cluster of the counties with the highest degree of involvement in the agricultural output, considering that they have not only the largest owned area and the biggest number of employees in the field of agriculture, but also the largest grain production.

The cluster 2016-3 consists of eight counties, namely Bacău, Botoşani, Iaşi, Mehedinţi, Mureş, Neamţ, Prahova, and Vaslui. The counties grouped into this cluster are characterized by the highest average value of the variable GDP/county-GDP.2016 (EUR 3311.31 million against the 2016 national average of EUR 2963.60 million) and the highest average value of the variable Working population in agriculture-POPAGR.2016 (49,925.00 workers in agriculture against the 2016 national average of 42,248.72 workers).

Moreover, the counties forming the cluster 2016-3 are the ones reporting the lowest average values of the variables Payments made through NRDP-NRDP.2016 (EUR 17,723.26 thousand against the 2016 national average of EUR 24,856.44 thousand) and Length of modernized public roads-ROADS.2016 (311.63 km of modernized district and communal roads against the 2016 national average of 423.79 km) but also the variable Fleet size of tractors and agricultural machinery-MACHINES.2016 (3986.00 tractors and agricultural machinery against the 2016 national average of 5020.69 tractors and agricultural machinery). As one can notice, the eight counties grouped into this cluster have not given much attention to attracting European funds, and in this respect, they were 28.7% below the 2016 national average and 74.9% below the average of the most competitive counties (grouped into the cluster 2016-4).

In cluster 2016-4, seven counties were grouped together out of the 39 kept for analysis: Bihor, Buzău, Galaţi, Giurgiu, Olt, Satu Mare, and Tulcea. These counties are defined not only by the highest average value of the variable Payments made through NRDP-NRDP.2016 (EUR 29,963.47 26 thousand against the 2016 national average of EUR 24,856.44 thousand), but also by the lowest average value of the variable GDP/county-GDP.2016 (EUR 2232.21 million against the 2016 national average of EUR 2963.60 million).

As concerns the clusters identified for 2020, they are presented in Table 13, alongside the main characteristics determined according to the variables included in the analysis.

The cluster 2020-1 is composed of 16 counties, as follows: Alba, Arges, Bacău, Buzău, Cluj, Constanţa, Dâmboviţa, Galaţi, Gorj, Iaşi, Mehedinţi, Neamţ, Prahova, Suceava, and Tulcea, alongside Vaslui.

The counties grouped into this cluster are characterized by the lowest average value of the variable Fleet size of tractors and agricultural machinery-MACHINES.2020 (4588.31 tractors and agricultural machinery against the 2020 national average of 5727.05 tractors and agricultural machinery), as well as by the lowest average value of the variable Agricultural output—grains-PRODAGR.2020 (329,038.56 71 tons of grain production against the 2020 national average of 423,793.28 tons of grain production).

In cluster 2020-2, five counties were grouped, that is, Arad, Bihor, Olt, Teleorman, and Timiş. These counties are defined by the highest average values for seven out of the eight analyzed variables, and this says a lot about their performance in attracting non-refundable funds and in investing them efficiently.

Hence, the counties grouped into the cluster 2020-2 register the highest average value of the variable Payments made through NRDP-NRDP.2020 (EUR 115,533.36 thousand against the national average registered in 2020 of EUR 70,577.30 thousand), the highest average value of the variable GDP/county-GDP.2020 (EUR 4812,13 million against the 2020 national average of EUR 3835.95 million), the highest average value of the variable EMPLOYEES.2020 (4337.60 employees in the field of agriculture against the 2020 national average of 3225.59 employees in agriculture), the highest average value of the variable Length of modernized public roads-ROADS.2020 (724.40 km of modernized district and communal roads against the 2020 national average of 551.49 km), the highest average value of the vari-

able Fleet size of tractors and agricultural machinery-MACHINES.2020 (8869.80 tractors and agricultural machinery against the 2020 national average of 5727.05 tractors and agricultural machinery), and the highest average value of the variable AREAS.2020 (354,771,20 ha of sown areas against the 2020 national average of 199,505.92 ha of sown areas), alongside the highest average value of the variable Working population in agriculture-POPAGR.2020 (51,660.00 workers in agriculture against the 2020 national average of 40,748.72 workers in agriculture).

**Table 13.** Cluster characteristics for 2020.

| Cluster | No. of Counties | Variables | Minimum | Maximum | Mean | Variance |
|---|---|---|---|---|---|---|
| Cluster 2020-1 | 16 | NRDP.2020 | 34,153.33 | 129,293.02 | 66,422.40 | 31,873.11 |
| | | GDP.2020 | 1638.15 | 10,865.13 | 4684.93 | 2865.28 |
| | | EMPLOYEES.2020 | 1585 | 4479 | 3152.69 | 917.84 |
| | | ROADS.2020 | 85 | 1236 | 612.94 | 338.79 |
| | | MACHINES.2020 | 2228 | 8199 | 4588.31 | 1849.78 |
| | | AREAS.2020 | 81,735 | 488,275 | 200,358.50 | 95,218.53 |
| | | PRODAGR.2020 | 219,347 | 468,156 | 329,038.56 | 64,161.19 |
| | | POPAGR.2020 | 21,600 | 72,200 | 45,600.00 | 12,668.18 |
| Cluster 2020-2 | 5 | NRDP.2020 | 49,812.12 | 170,254.02 | 115,533.36 | 43,928.60 |
| | | GDP.2020 | 2122.66 | 9737.54 | 4812.13 | 3002.19 |
| | | EMPLOYEES.2020 | 3065 | 6026 | 4337.60 | 1065.78 |
| | | ROADS.2020 | 321 | 874 | 724.40 | 228.06 |
| | | MACHINES.2020 | 6505 | 11,996 | 8869.80 | 2510.00 |
| | | AREAS.2020 | 311,045 | 413,814 | 354,771.20 | 41,555.26 |
| | | PRODAGR.2020 | 952,194 | 1,325,852 | 1,088,111.00 | 153,328.09 |
| | | POPAGR.2020 | 34,100 | 60,300 | 51,660.00 | 10,462.46 |
| Cluster 2020-3 | 11 | NRDP.2020 | 29,302.23 | 150,340.50 | 57,786.58 | 34,734.38 |
| | | GDP.2020 | 1567.09 | 7284.01 | 3093.72 | 1660.90 |
| | | EMPLOYEES.2020 | 1728 | 4098 | 2637.45 | 704.88 |
| | | ROADS.2020 | 63 | 1135 | 476.27 | 327.25 |
| | | MACHINES.2020 | 2811 | 10,838 | 6343.09 | 2190.25 |
| | | AREAS.2020 | 51,491 | 120,139 | 68,723.36 | 18,983.00 |
| | | PRODAGR.2020 | 68,369 | 213,370 | 129,554.55 | 39,411.19 |
| | | POPAGR.2020 | 17,500 | 50,100 | 30,654.55 | 10,114.28 |
| Cluster 2020-4 | 7 | NRDP.2020 | 42,639.99 | 110,870.92 | 68,062.45 | 23,525.93 |
| | | GDP.2020 | 1464.41 | 4622.96 | 2364.50 | 1077.17 |
| | | EMPLOYEES.2020 | 1802 | 5354 | 3522.14 | 1336.78 |
| | | ROADS.2020 | 138 | 681 | 405.71 | 227.52 |
| | | MACHINES.2020 | 2148 | 7669 | 5117.00 | 1941.19 |
| | | AREAS.2020 | 185,500 | 386,096 | 292,168.86 | 71,900.33 |
| | | PRODAGR.2020 | 462,008 | 710,482 | 628,238.00 | 82,798.59 |
| | | POPAGR.2020 | 27,500 | 52,100 | 37,728.57 | 9007.54 |

Source: calculations made by the authors.

The cluster 2020-3 consists of 11 counties, as follows: Bistriţa-Năsăud, Brăila, Călăraşi, Covasna, Harghita, Hunedoara, Maramureş, Satu Mare, Sibiu, Vâlcea, and Vrancea. The thing about these counties is that they register more minimum values of the averages of the selected variables, that is, for five out of the eight analyzed variables.

We can observe that the counties grouped into the cluster 2020-3 registered the lowest average value of the variable Payments made through NRDP-NRDP.2020 (EUR 57,786.58 thousand against the national average of EUR 70,577.30 thousand reported in 2020, which represents less than half of the average value), the lowest average value of the variable EMPLOYEES.2020 (2637.45 employees in agriculture against the 2020 average value of 3225.59 employees in agriculture), the lowest average value of the variable AREAS.2020 (68,723.36 ha of sown areas against the 2020 national average of 199,505.92 ha of sown areas), the lowest average value of the variable Agricultural output—grains-PRODAGR.2020 (129,554.55 tons of grain production against the 2020 national average of 423,793.28 tons of grain production), and last but not least, the lowest average value for the variable POAGR.2020 (30,654.55 employees in agriculture against the 2020 national average of 40,748.72 employees in agriculture).

In the cluster 2020-4, seven counties were joined together for the analysis, namely Botoşani, Braşov, Caraş-Severin, Giurgiu, Ialomiţa, Mureş, and Sălaj. These counties are characterized by two minimum values, that is, the lowest average value of the variable GDP/county-GDP.2020 (EUR 2364.50 million against the 2020 national average of EUR 3835.95 million), alongside the lowest average value of the variable Length of modernized public roads-ROADS.2020 (405.71 km of modernized district and communal roads against the 2020 national average of 551.49 km).

## 5. Conclusions

The rural development policy is a new political field of the EU which aspires to integrally tackle a large area of issues and development sectors at different territorial levels. New forms of governance have been developed which are characterized by decentralization, partnership, participation, and new formal mechanisms of the horizontal and vertical coordination of policies. Although the EU's rural development policy defines a new set of policies for older components, it seems to go beyond those borders and transform itself in a policy domain which is potentially distinct. Nevertheless, until today this promising domain has been overshadowed by the CAP, both as it regards financing and as it concerns political decision making.

The issue surrounding the financing of rural areas, with regard to both EU member states and Romania, is one of the key problems which at present the governmental and regional authorities must prioritize because it is well known that rural areas and their long-term development level affect economic, social, and environmental sustainability, while the financing and efficiency of the activities carried out in these regions are closely linked with the effectiveness of financial resources allotment.

In order to be efficient and have a high yield, the financing of different investment objectives from European funds must be channeled through several new economic concepts, such as "innovative potential", "smart village", "multifunctionality", "multisectoral approach", "social return on investment", and "territorial justice", which can identify the real and specific problems of each rural region so that the EU's rural development policy within each member state will become at the same time a driving force behind the increase in both the population's standard of living and the continuous development of the member states' rural areas.

Romania, having become a member state of the EU (on 1 January 2007) is still in the process of rural development in a new framework, falling into step with the principles, objectives, and mechanisms of rural development of this integrationist organization. Our country has coordinated the National Rural Development Program with the European strategies and programs in the field, making it an integrative part of these strategies and programs and enjoying adequate responsibilities, contributions, and benefits.

The fulfillment of the general objective of the research depended on setting and attaining certain goals specific to the field, such as:

- pointing out the issues of the CAP, particularly the second pillar—the rural development—with which our country has to be permanently in connection;

- analysis of the objectives of rural development and of the strategies elaborated to accomplish them;
- highlighting the essential characteristics of the new stage for the rural development process in our country, that is, the Rural Development Program;
- structuring a statistical analysis framework to identify the relevant infrastructure for rural development within Romania's development regions;
- analysis of the economic and social impact of the European funds for development on the rural areas in Romania by using econometric research.

Hence, in order to obtain relevant results in relation to the goal pursued, we used the hierarchical cluster analysis for observing the effects of payments made through the NRDP on the development of rural areas. The hierarchical cluster analysis allowed us to identify groups of variables with similar characteristics and, as a consequence, we quantified the structural characteristics of the samples or variables selected for analysis. The procedure of hierarchical grouping involves building a structural hierarchy. The cluster analysis served to develop decision-making rules and then to apply these rules in order to attribute a heterogeneous collection of variables to a series of linked subsets of data (clusters). This is a practical method, rather than a theoretical one, for exploring the relations that emerged between the analyzed variables. The final result of our cluster is a graphical interface of classifications and a set of decision-making rules to allocate new variables within the obtained classifications.

In order to explore the impact of the European funds on the rural development in Romania, we analyzed the following variables grouped round the counties (NUTS 3) for the years 2016 and 2020: payments made through NRDP, GDP/county, number of employees in agriculture, length of modernized county and communal roads, size of the tractor and machinery fleet, sown areas of main crops, and agricultural output—grains as well as the working population in agriculture.

By putting together the results obtained in our study we can state the obvious; that is, significant progress was registered in only 4 years, and it is clear that this progress could not have been possible without a massive investment program.

If we look at the evolution of the annual average values for the analyzed interval, we can notice that the GDP rose by 29.44% (from an average value/county of EUR 2963.60 million in 2016 up to EUR 3835.95 million in 2020). The number of employees in the fields of agriculture, forestry, and fishing registered an increase of 10.88% with respect to the analyzed interval, but, on the other hand, we can notice a drop of 3.56% of the working population in these sectors. This evolution is a normal one which validates the evolution of the two variables at stake, given the fact that we can notice the statistical transfer of a number of persons who declare agriculture as their occupation towards the rise in the number of employees in the field of agriculture, with positive effects on both the individual and the economic welfare. This process is very likely due to the projects financed by the European funds, which require the existence of labor contracts for different agricultural activities involving the measures supporting the development of rural areas.

As concerns the length of modernized district and rural roads, we can see a 30.13% increase in the average modernized length, from 423.79 km/county in 2016 up to 551.49 km/county in 2020. Most likely, it is not by chance that the five counties grouped into the 2020-2 cluster which have attracted the highest average amounts of NRDP funds are also the ones with the biggest average length of modernized district and rural roads (724.40 km, 31% more than the national average).

Similarly, the analysis regarding the fleet of tractors and agricultural machinery points out a 14.07% increase in the average number/county, from 5020.69 tractors and agricultural machinery in 2016 to 5727.05 tractors and agricultural machinery in 2020). Just like in the case of the number of kilometers of modernized district and rural roads, with regard to the number of tractors and agricultural machinery, we can see that the highest average value is concentrated within the counties grouped in the 2020-2 cluster (that is to say, 8869.80 tractors and agricultural machinery, 54% more than the national average); further-

more, these are counties which have been superior in terms of attracting and efficiently spending the European funds made available for the beneficiaries of the NRDP.

By examining the variables regarding both the sown area of main crops and the agricultural output—grains—we can see a slight drop in the values, and this is not very easy to interpret. On the one hand, the sown area of main crops decreased marginally by 1.64% in 2020 compared to 2016. If we can agree upon the fact that a small part of the cultivated fields changed their destination (because of the different measures stipulated in the NRDP), it is very hard to believe that the agricultural production was influenced to such an extent. In this regard, the most likely plausible explanation is given by the fact that it was not a good agricultural year which generated a lower vegetal agricultural production.

In conclusion, as a result of our analysis we can say that a series of positive effects can be highlighted thanks to the existence of the non-refundable European funds which trigger positive direct and indirect effects on the overall economy and the population's standard of living. Moreover, certain positive influences of the non-refundable European funds can be spotted if other econometrical methods are put into practice to quantify their effects, thus opening the way for future thorough research.

In our study, we encountered both internal limits specific to an extremely complex and interdisciplinary analysis and limits which resulted from the theoretical, qualitative, and quantitative research, generated in its turn by certain limitations of the databases, such as the existence of incomplete data on the national and European financial allocation for rural development in Romania (measures 12 and 18, financed by the NRDP).

With regard to the potential subsequent development, we propose that the research will be continued by the analysis of the strategic priorities of financing with respect to the intelligent development of rural areas and rural towns and the implementation of innovatory concepts of rural development, as well as their integration into the long-term projects of environmental protection projects.

**Author Contributions:** Formal analysis, R.P. and L.D.; investigation, B.B., R.-Ș.B., S.D. and L.Ț.; methodology, L.Ț. and S.D.; supervision, R.P. and B.B.; validation, L.D. and R.-Ș.B.; writing—original draft, R.P. and L.Ț.; writing—review and editing, L.D. and S.D. All authors have read and agreed to the published version of the manuscript.

**Funding:** This work was supported by the grant POCU380/6/13/123990, co-financed by the European Social Fund within the Sectorial Operational Program Human Capital 2014–2020.

**Institutional Review Board Statement:** Not applicable.

**Informed Consent Statement:** Not applicable.

**Data Availability Statement:** Not applicable.

**Conflicts of Interest:** The authors declare no conflict of interest.

## Appendix A

**Table A1.** Cluster agglomeration table for the year 2016.

| Stage | Cluster Combined | | Coefficients | Stage Cluster First Appears | | Next Stage |
|---|---|---|---|---|---|---|
| | Cluster 1 | Cluster 2 | | Cluster 1 | Cluster 2 | |
| 1 | 4 | 28 | 117,300,663.937 | 0 | 0 | 20 |
| 2 | 15 | 31 | 257,832,789.792 | 0 | 0 | 5 |
| 3 | 26 | 30 | 407,476,050.192 | 0 | 0 | 20 |
| 4 | 21 | 22 | 584,498,719.964 | 0 | 0 | 8 |

**Table A1.** *Cont.*

| Stage | Cluster Combined | | Coefficients | Stage Cluster First Appears | | Next Stage |
|---|---|---|---|---|---|---|
| | Cluster 1 | Cluster 2 | | Cluster 1 | Cluster 2 | |
| 5 | 15 | 33 | 823,346,906.760 | 2 | 0 | 23 |
| 6 | 3 | 16 | 1,126,752,163.873 | 0 | 0 | 19 |
| 7 | 12 | 38 | 1,484,144,701.797 | 0 | 0 | 15 |
| 8 | 6 | 21 | 1,935,538,761.009 | 0 | 4 | 14 |
| 9 | 9 | 25 | 2,474,762,090.325 | 0 | 0 | 14 |
| 10 | 14 | 17 | 3,057,669,357.855 | 0 | 0 | 29 |
| 11 | 8 | 23 | 3,651,047,716.609 | 0 | 0 | 25 |
| 12 | 1 | 40 | 4,250,147,576.482 | 0 | 0 | 17 |
| 13 | 18 | 37 | 5,031,190,707.834 | 0 | 0 | 21 |
| 14 | 6 | 9 | 5,840,226,597.818 | 8 | 9 | 23 |
| 15 | 12 | 20 | 6,685,034,516.450 | 7 | 0 | 28 |
| 16 | 19 | 32 | 7,611,566,621.605 | 0 | 0 | 27 |
| 17 | 1 | 13 | 8,691,913,465.661 | 12 | 0 | 19 |
| 18 | 7 | 39 | 9,857,468,392.247 | 0 | 0 | 26 |
| 19 | 1 | 3 | 11,804,739,843.778 | 17 | 6 | 32 |
| 20 | 4 | 26 | 13,758,812,766.139 | 1 | 3 | 30 |
| 21 | 10 | 18 | 16,088,626,074.555 | 0 | 13 | 27 |
| 22 | 24 | 27 | 18,552,624,981.333 | 0 | 0 | 26 |
| 23 | 6 | 15 | 22,140,533,434.261 | 14 | 5 | 35 |
| 24 | 5 | 29 | 28,206,653,271.581 | 0 | 0 | 33 |
| 25 | 2 | 8 | 34,521,357,875.169 | 0 | 11 | 31 |
| 26 | 7 | 24 | 41,437,246,703.230 | 18 | 22 | 30 |
| 27 | 10 | 19 | 49,315,657,250.408 | 21 | 16 | 33 |
| 28 | 12 | 34 | 57,577,115,126.121 | 15 | 0 | 32 |
| 29 | 11 | 14 | 72,413,549,371.897 | 0 | 10 | 34 |
| 30 | 4 | 7 | 91,885,142,012.891 | 20 | 26 | 36 |
| 31 | 2 | 35 | 114,176,624,493.390 | 25 | 0 | 34 |
| 32 | 1 | 12 | 141,418,393,238.795 | 19 | 28 | 35 |
| 33 | 5 | 10 | 186,328,292,651.205 | 24 | 27 | 36 |
| 34 | 2 | 11 | 247,183,216,021.678 | 31 | 29 | 38 |
| 35 | 1 | 6 | 376,575,212,742.441 | 32 | 23 | 37 |
| 36 | 4 | 5 | 606,216,284,565.000 | 30 | 33 | 37 |
| 37 | 1 | 4 | 1,773,581,359,591.390 | 35 | 36 | 38 |
| 38 | 1 | 2 | 5,802,040,263,977.120 | 37 | 34 | 0 |

Source: calculations made by the authors.

**Appendix B**

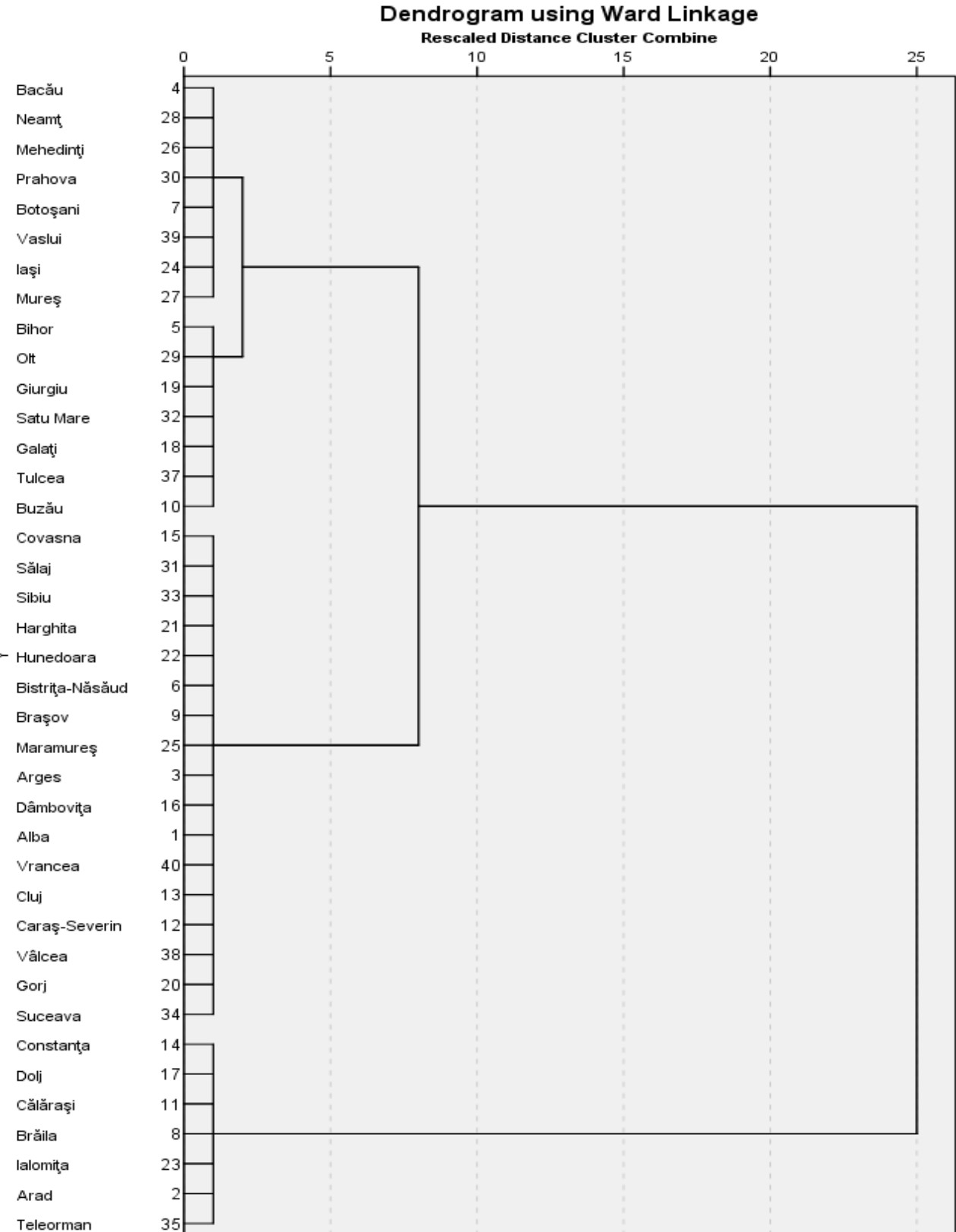

**Figure A1.** Cluster dendrogram for the year 2016. Source: calculations made by the authors.

## Appendix C

**Table A2.** Results of the dispersion homogeneity test for the year 2016.

|  |  | Levene Statistic | df1 | df2 | Sig. |
|---|---|---|---|---|---|
| NRDP.2016 | Based on Mean | 5.965 | 3 | 35 | 0.102 |
|  | Based on Median | 1.786 | 3 | 35 | 0.168 |
|  | Based on Median and with adjusted df | 1.786 | 3 | 23.630 | 0.177 |
|  | Based on trimmed mean | 5.442 | 3 | 35 | 0.074 |
| GDP.2016 | Based on Mean | 1.207 | 3 | 35 | 0.322 |
|  | Based on Median | 0.716 | 3 | 35 | 0.549 |
|  | Based on Median and with adjusted df | 0.716 | 3 | 26.622 | 0.551 |
|  | Based on trimmed mean | 1.117 | 3 | 35 | 0.355 |
| EMPLOYEES.2016 | Based on Mean | 0.875 | 3 | 35 | 0.463 |
|  | Based on Median | 0.418 | 3 | 35 | 0.741 |
|  | Based on Median and with adjusted df | 0.418 | 3 | 33.001 | 0.741 |
|  | Based on trimmed mean | 0.784 | 3 | 35 | 0.511 |
| ROADS.2016 | Based on Mean | 1.520 | 3 | 35 | 0.226 |
|  | Based on Median | 0.708 | 3 | 35 | 0.554 |
|  | Based on Median and with adjusted df | 0.708 | 3 | 22.575 | 0.557 |
|  | Based on trimmed mean | 1.371 | 3 | 35 | 0.268 |
| MACHINES.2016 | Based on Mean | 3.203 | 3 | 35 | 0.035 |
|  | Based on Median | 1.371 | 3 | 35 | 0.268 |
|  | Based on Median and with adjusted df | 1.371 | 3 | 17.691 | 0.284 |
|  | Based on trimmed mean | 3.065 | 3 | 35 | 0.061 |
| AREAS.2016 | Based on Mean | 0.704 | 3 | 35 | 0.556 |
|  | Based on Median | 0.248 | 3 | 35 | 0.862 |
|  | Based on Median and with adjusted df | 0.248 | 3 | 30.190 | 0.862 |
|  | Based on trimmed mean | 0.644 | 3 | 35 | 0.592 |
| PRODAGR.2016 | Based on Mean | 2.593 | 3 | 35 | 0.068 |
|  | Based on Median | 1.885 | 3 | 35 | 0.150 |
|  | Based on Median and with adjusted df | 1.885 | 3 | 23.844 | 0.159 |
|  | Based on trimmed mean | 2.582 | 3 | 35 | 0.069 |
| POPAGR.2016 | Based on Mean | 1.159 | 3 | 35 | 0.339 |
|  | Based on Median | 0.478 | 3 | 35 | 0.700 |
|  | Based on Median and with adjusted df | 0.478 | 3 | 27.495 | 0.700 |
|  | Based on trimmed mean | 1.049 | 3 | 35 | 0.383 |

Source: calculations made by the authors.

## Appendix D

**Table A3.** Cluster agglomeration table for the year 2020.

| Stage | Cluster Combined | | Coefficients | Stage Cluster First Appears | | Next Stage |
|---|---|---|---|---|---|---|
| | **Cluster 1** | **Cluster 2** | | **Cluster 1** | **Cluster 2** | |
| 1 | 31 | 33 | 64,404,792.244 | 0 | 0 | 2 |
| 2 | 12 | 31 | 240,763,597.486 | 0 | 1 | 7 |
| 3 | 3 | 30 | 461,728,831.261 | 0 | 0 | 5 |
| 4 | 22 | 25 | 978,237,168.306 | 0 | 0 | 6 |
| 5 | 3 | 28 | 1,675,339,338.792 | 3 | 0 | 23 |
| 6 | 9 | 22 | 2,393,961,089.644 | 0 | 4 | 13 |
| 7 | 12 | 15 | 3,389,591,367.057 | 2 | 0 | 19 |
| 8 | 10 | 24 | 4,528,926,696.027 | 0 | 0 | 28 |
| 9 | 1 | 20 | 5,734,474,108.876 | 0 | 0 | 14 |
| 10 | 18 | 37 | 6,945,606,374.357 | 0 | 0 | 26 |
| 11 | 8 | 11 | 8,232,490,216.027 | 0 | 0 | 31 |
| 12 | 34 | 39 | 9,743,869,524.562 | 0 | 0 | 26 |
| 13 | 9 | 21 | 11,425,216,376.040 | 6 | 0 | 24 |
| 14 | 1 | 13 | 13,498,578,501.613 | 9 | 0 | 23 |
| 15 | 4 | 16 | 15,748,608,417.152 | 0 | 0 | 22 |
| 16 | 2 | 29 | 18,219,237,672.362 | 0 | 0 | 21 |
| 17 | 19 | 32 | 20,811,288,321.837 | 0 | 0 | 18 |
| 18 | 19 | 27 | 23,624,290,230.649 | 17 | 0 | 31 |
| 19 | 12 | 40 | 26,782,831,146.754 | 7 | 0 | 20 |
| 20 | 12 | 38 | 31,108,938,837.629 | 19 | 0 | 27 |
| 21 | 2 | 5 | 36,324,058,747.482 | 16 | 0 | 34 |
| 22 | 4 | 26 | 41,874,416,155.565 | 15 | 0 | 29 |
| 23 | 1 | 3 | 47,759,667,893.552 | 14 | 5 | 29 |
| 24 | 6 | 9 | 54,845,345,479.988 | 0 | 13 | 27 |
| 25 | 7 | 23 | 62,619,699,686.954 | 0 | 0 | 32 |
| 26 | 18 | 34 | 71,117,882,101.408 | 10 | 12 | 28 |
| 27 | 6 | 12 | 84,700,008,054.715 | 24 | 20 | 36 |
| 28 | 10 | 18 | 102,343,663,128.080 | 8 | 26 | 33 |
| 29 | 1 | 4 | 122,346,503,283.710 | 23 | 22 | 33 |
| 30 | 35 | 36 | 142,482,381,440.997 | 0 | 0 | 34 |
| 31 | 8 | 19 | 168,373,392,992.570 | 11 | 18 | 32 |
| 32 | 7 | 8 | 204,015,903,659.138 | 25 | 31 | 37 |
| 33 | 1 | 10 | 259,410,452,092.336 | 29 | 28 | 35 |
| 34 | 2 | 35 | 340,756,938,440.127 | 21 | 30 | 37 |
| 35 | 1 | 14 | 433,056,152,499.026 | 33 | 0 | 36 |
| 36 | 1 | 6 | 807,385,479,003.819 | 35 | 27 | 38 |
| 37 | 2 | 7 | 1,442,841,519,344.310 | 34 | 32 | 38 |
| 38 | 1 | 2 | 4,411,531,227,086.600 | 36 | 37 | 0 |

Source: calculations made by the authors.

**Appendix E**

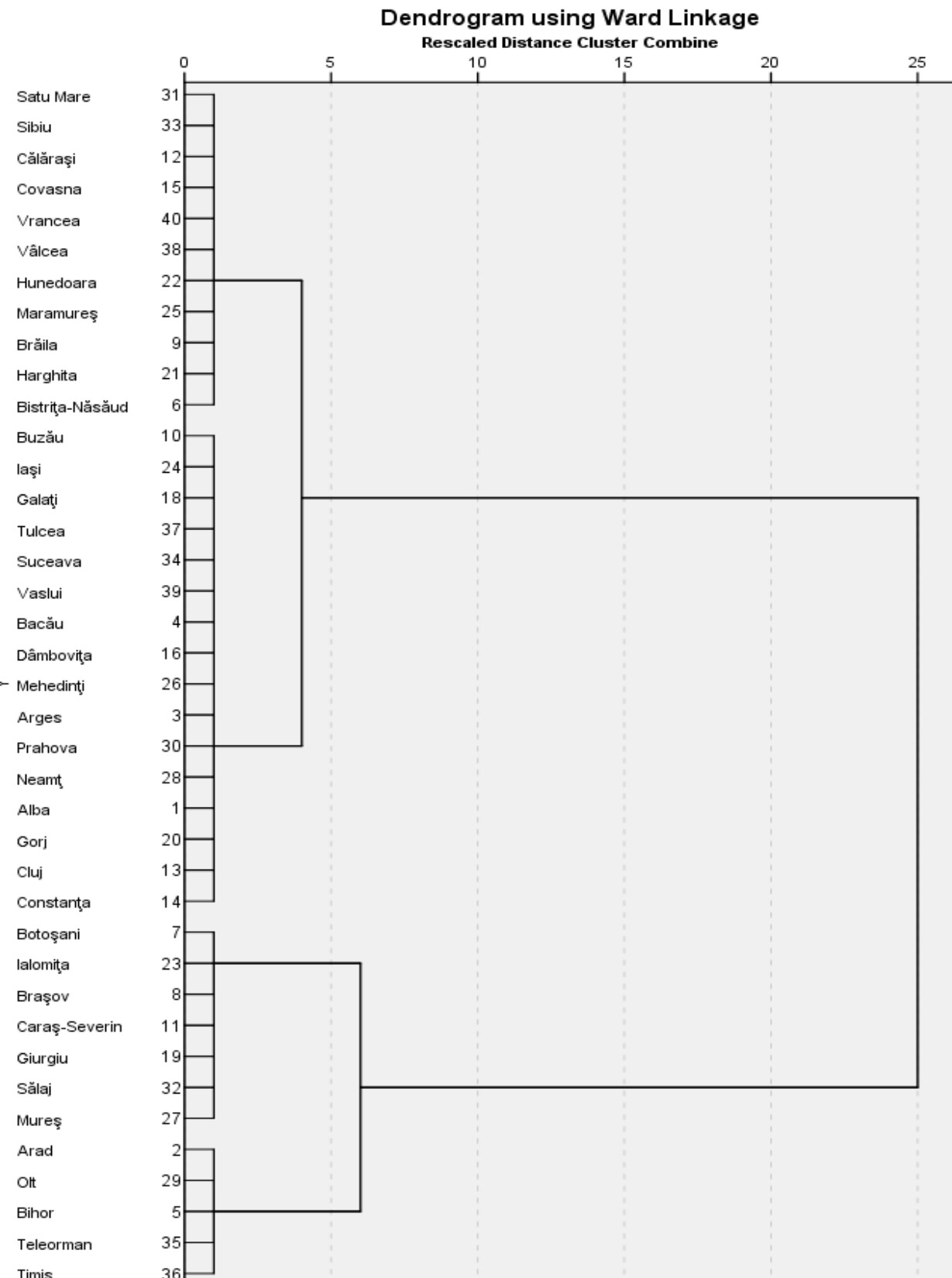

**Figure A2.** Cluster dendrogram for the year 2020. Source: calculations made by the authors.

## Appendix F

**Table A4.** Results of the dispersion homogeneity test for the year 2020.

|  |  | Levene Statistic | df1 | df2 | Sig. |
|---|---|---|---|---|---|
| NRDP.2020 | Based on Mean | 0.499 | 3 | 35 | 0.685 |
|  | Based on Median | 0.247 | 3 | 35 | 0.863 |
|  | Based on Median and with adjusted df | 0.247 | 3 | 31.523 | 0.863 |
|  | Based on trimmed mean | 0.452 | 3 | 35 | 0.718 |
| GDP.2020 | Based on Mean | 2.677 | 3 | 35 | 0.062 |
|  | Based on Median | 1.091 | 3 | 35 | 0.366 |
|  | Based on Median and with adjusted df | 1.091 | 3 | 27.696 | 0.370 |
|  | Based on trimmed mean | 2.322 | 3 | 35 | 0.092 |
| EMPLOYEES.2020 | Based on Mean | 1.982 | 3 | 35 | 0.135 |
|  | Based on Median | 1.168 | 3 | 35 | 0.336 |
|  | Based on Median and with adjusted df | 1.168 | 3 | 26.939 | 0.340 |
|  | Based on trimmed mean | 2.011 | 3 | 35 | 0.130 |
| ROADS.2020 | Based on Mean | 0.529 | 3 | 35 | 0.666 |
|  | Based on Median | 0.689 | 3 | 35 | 0.565 |
|  | Based on Median and with adjusted df | 0.689 | 3 | 31.122 | 0.566 |
|  | Based on trimmed mean | 0.558 | 3 | 35 | 0.646 |
| MACHINES.2020 | Based on Mean | 0.451 | 3 | 35 | 0.718 |
|  | Based on Median | 0.145 | 3 | 35 | 0.932 |
|  | Based on Median and with adjusted df | 0.145 | 3 | 29.503 | 0.932 |
|  | Based on trimmed mean | 0.443 | 3 | 35 | 0.724 |
| AREAS.2020 | Based on Mean | 3.671 | 3 | 35 | 0.061 |
|  | Based on Median | 2.174 | 3 | 35 | 0.109 |
|  | Based on Median and with adjusted df | 2.174 | 3 | 20.036 | 0.123 |
|  | Based on trimmed mean | 3.211 | 3 | 35 | 0.075 |
| PRODAGR.2020 | Based on Mean | 5.213 | 3 | 35 | 0.084 |
|  | Based on Median | 2.195 | 3 | 35 | 0.106 |
|  | Based on Median and with adjusted df | 2.195 | 3 | 13.595 | 0.135 |
|  | Based on trimmed mean | 4.819 | 3 | 35 | 0.097 |
| POPAGR.2020 | Based on Mean | 0.170 | 3 | 35 | 0.916 |
|  | Based on Median | 0.236 | 3 | 35 | 0.871 |
|  | Based on Median and with adjusted df | 0.236 | 3 | 33.009 | 0.871 |
|  | Based on trimmed mean | 0.185 | 3 | 35 | 0.906 |

Source: calculations made by the authors.

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
