# Peer review of "The Impact of RDP Measures on the Rural Development: The Case of Romania"

_sustainability, doi:10.3390/su14084857_

Round 1

Reviewer 1 Report

Overall, the work is of a good standard. I suggest the following revisions:

  - 3. Research Methodology p. 6 III paragraph: Initial sentence repeated twice;

- the final Appendices are not reflected in the text; I suggest to modify to understand the insertion;

- in the final comments and in the discussion, I suggest to better explain the methodology and the results so that they can both be understandable for a better scientific dissemination even outside the specific disciplinary sector.

Reviewer 2 Report

Dear author:
For rural economically poor areas, rural reconstruction programs can improve this problem. It is of great help in changing the predicament of rural economy and community development. The author explores this issue, which will once again promote the economic growth of local rural areas. However, the manuscript will need to be reworked to revise the narrative and follow social science research writing requirements to reinforce the importance of the manuscript. Therefore, the following suggestions will be made for the author's reference.
It is recommended that the author submit a new manuscript after improving the relevant suggestions, which will help readers understand the importance and value of this manuscript.

1. About the introduction
a. Sentences in some paragraphs are too long. Such as: " The rural develo..." ; " Hence, because of the extremely changing geopolitical..."
b. The purpose is not clearly stated. Please adjust the narration.
c. It is suggested to delete this paragraph, which is not related to the subject of the manuscript. "This paper is divided into five sections, starting with an..."

2. About Literature Review
The research is not a literature review type of research, and the purpose of this chapter does not seem easy to understand the value of existence.
Moreover, the meaning of literature review or discussion is usually to define the main key terms of the manuscript, and to discuss the significance of the relevant research results to the topic.
a. Please clearly define the CAP.
b. The current state of decision-making in Romanian RDP?
c. What key points did you see in the literature discussion?
I actually have a hard time understanding the meaning of this chapter.
Can it be improved.
Recommendations are:
a. The first part states that the decision of Romanian RDP (this includes CAP)
b. The direction of EU-related decisions
c. Analysis of the research and development status of RDP decision-making
d. An introduction to hierarchical cluster analysis and a summary of the authors' rationale

3. About research methods
a.N, what does 39 mean?
Why is the sample size only 39? Does this meet the basic requirements of research data? Please explain and suggest supporting evidence.
b. Under the table, please mark the standard value of "*", *p<0.05 .

4. Discussion and conclusion section
After the author's analysis, the results obtained should be explained in the discussion section.
The conclusion is the answer obtained by summarizing the results of the research. References should no longer be included. Instead, it is necessary to present the researcher's opinion after investigation and analysis.
Suggest the appropriateness of restating or arranging the content of the paragraph.

This is an interesting topic that could help improve local rural development decision-making in Romania, but the manuscript still has too many flaws and the authors are advised to submit it with substantial adjustments. 

Reviewer 3 Report

1. There are many problems with language and grammar, such as our goal is to In accordance with the methodological approach of research; RibašauskienË™e et al. [40]; et. al [38]. Please check the full text carefully.
2. The method used in this paper needs to be introduced in detail, and the current analysis is not clear.
3. Explain the applicability of the method.
4. Why only study 2016 and 2020, and what about other years?
5. Should the data in Table 4, Table 5, Table 7, Table 8 and Table 10 be explained in detail? For example: Statistic, df1, df2 and Sig.

Reviewer 4 Report

 The Impact of RDP Measures on the Rural Development: The Case of Romania

The current manuscript is written and presented with details in the research steps and results. Some minor points are required to improve or clarify.

1.There is no research hypotheses constructed and empirically tested in this paper. It will be more rigorous if research hypotheses are constructed from theories and/or existing literature.

2. There is in section 3 Research Methodology a phrase that is repeated:

3. In the methodology of statistical testing the rule of decisions, in conditions of risk assumed, is for reject null hypothesis. If you want to specify the rule for accept the null hypothesis you must specify the risk for type II errors. (Page 10, equation 4)

4. There are a lot of notations on pages 9 and 10, please specify what means this.

5. Please present the argumentations of using Welch AND (not or) Brown-Forsyte test instead of Levene test and ANOVA.

6. In the section 4 Empirical Results and Discussion please replace the symbols of the variables with the names of variables for a better understanding of results.

7. Conclusion part seems too weak. As mentioned, research objectives clarified could help here with emphasizing most important implications and revealing the contribution of this research to the scientific knowledge.

8. In the References (16, 27) please remove the phrase “indexata în 17 baze de date printre care: EconPapers, IDEAS, REPEC, EBSCO, DOAJ".

Round 2

Reviewer 2 Report

Dear author
On the basis of the revised manuscript, the author has made efforts and achieved good results.
But regarding the presentation of the numbers in Tables 7, 8, 9, 10, 11, 12, I am puzzled.
For example, in Table 7, the value of df2 is presented, 14,066, I can understand that he is the value of 14,066. In Sig. value, 0,037 means, 0.037 ? or 0,037 ?
In numerical representation, " . " and " , " have different meanings.
The author must confirm.

I hope this issue needs to be resolved in order to present clearer content and qualify for submission.
wish all the best

Author Response

We appreciate the time and effort that you dedicated to providing feedback on our manuscript and are grateful for the insightful comments on and valuable improvements to our paper. We believe these revisions have resulted in a significantly improved manuscript. 

Thank you for the helpful comments!

Reviewer 3 Report

I'm satisfied by the responses of authors!

Author Response

Thank you for the helpful comments!
